# Acute Effects of Air Pollution and Noise from Road Traffic in a Panel of Young Healthy Adults

**DOI:** 10.3390/ijerph16050788

**Published:** 2019-03-04

**Authors:** Hanns Moshammer, Julian Panholzer, Lisa Ulbing, Emanuel Udvarhelyi, Barbara Ebenbauer, Stefanie Peter

**Affiliations:** ZPH, Environmental Health, Medical University of Vienna, 1090 Vienna, Austria; julian.panholzer@gmail.com (J.P.); lisa.ulbing@meduniwien.ac.at (L.U.); emanuel.udvarhelyi@gmail.com (E.U.); n0611852@students.meduniwien.ac.at (B.E.); n1208446@students.meduniwien.ac.at (S.P.)

**Keywords:** road traffic, air pollution, noise, physiological effects, panel study, healthy young adults

## Abstract

Panel studies are an efficient means to assess short-term effects of air pollution and other time-varying environmental exposures. Repeated examinations of volunteers allow for an in-depth analysis of physiological responses supporting the biological interpretation of environmental impacts. Twenty-four healthy students walked for 1 h at a minimum of four separate occasions under each of the following four settings: along a busy road, along a busy road wearing ear plugs, in a park, and in a park but exposed to traffic noise (65 dB) through headphones. Particle mass (PM_2.5_, PM_1_), particle number, and noise levels were measured throughout each walk. Lung function and exhaled nitrogen oxide (NO) were measured before, immediately after, 1 h after, and approximately 24 h after each walk. Blood pressure and heart rate variability were measured every 15 min during each walk. Recorded air pollution levels were found to correlate with reduced lung function. The effects were clearly significant for end-expiratory flows and remained visible up to 24 h after exposure. While immediate increases in airway resistance could be interpreted as protective (muscular) responses to particulate air pollution, the persisting effects indicate an induced inflammatory reaction. Noise levels reduced systolic blood pressure and heart rate variability. Maybe due to the small sample size, no effects were visible per specific setting (road vs. park).

## 1. Introduction

Acute effects of air pollutants can be examined by time series studies [1,2,3,4,5,6] or panel studies [7,8]. Both approaches correlate temporal changes in exposure markers to temporal changes in health endpoints. The former type of studies focuses on population level effects and usually uses data obtained from health registries (e.g., hospital admission [1,3], outpatient disease events [6], or mortality data [1,2,4,5]). The latter type concentrates on individual participants. The disadvantage of a typically smaller number of participants in this type of studies is often compensated by more detailed information obtained by repeated measurements [9,10,11] or through detailed symptom diaries kept by the participants [12]. The more complicated the physiological measurements, the higher the effort per person. Therefore, panel studies with the requirement of more invasive or demanding examinations often rely on fewer participants. Previous panel studies on health effects of air pollution have relied on numbers of participants ranging from 13 to 163 [12,13,14,15].

Bloemsma et al. [7] provide an overview on 25 panel studies on the acute effects of air pollution in patients with chronic obstructive pulmonary disease (COPD), published between 1993 and February 2016. Participant numbers varied between 16 and 459. In studies with fewer participants, repeated lung function tests or analyzed blood samples for inflammatory markers were performed, while the larger studies relied on symptom diaries, reported reasons for consultation of physicians, or reports about limitation in activities.

Acute health effects of short-term air pollution episodes appear to be small when compared to the effects of prolonged or chronic exposure to similar concentrations of air pollution [16]. From a public health point of view, chronic exposure and longer averaging periods of exposure to pollution are more relevant than rare peak episodes. Nevertheless, the analysis of health effects of temporal variation in pollution concentration is important as it supports the causal interpretation of epidemiological findings in cohort studies. Confounding factors may differ between acute and chronic effects. If the same endpoints are demonstrated to be affected in both types of studies, this serves as a strong argument against a possible systematic confounding.

Panel studies can be enhanced by randomly [17,18] or systematically [19,20,21,22] placing participants in various settings that differ in exposure levels. In that case, either natural settings with different pollution exposures are sought or experimental exposure settings are provided. Natural settings clearly have the benefit of a direct relevance for everyday life. Under realistic conditions, air pollution is not a binary (yes/no) variable, but there is always an exposure of a variable magnitude. Analyzing the effects using a simplified binary (high/low) exposure variable reduces study power, compared to the use of continuous variables where available [23]. Therefore, switching settings may be favorable for augmenting the exposure contrast, but not for solely defining the exposure variable.

The effects of acute air pollution have been repeatedly demonstrated by panel studies. In these investigations, COPD patients [9,10,11,13,14,24], asthma patients [12,15,25,26,27,28,29], and patients with metabolic [30] or cardiovascular diseases [31,32] (e.g., patients with dual chamber implantable cardioverter-defibrillators [33] or with hypertension [34]) were examined. Such panel studies have been performed in the elderly [35,36,37,38] and in areas of high air pollution [39,40,41,42,43,44], but less so in young and healthy adults in settings with moderate to low air pollution levels [45,46,47,48,49,50,51]. Patients with respiratory diseases were usually examined for respiratory effects like lung function changes or related symptoms, while patients with cardiovascular and metabolic diseases were checked for cardiovascular and inflammatory parameters. The few studies in healthy adults did investigate very different endpoints, ranging from inflammatory markers [46,49] to heart-rate variability [45] and arterial stiffness [48,49]. Only a few focused on lung function [45,46,50,51], and these applied very different averaging times, ranging from one hour [50,51] to one day or even 2 weeks [46], and investigated different lag periods.

Noise pollution is a major concern for the modern world, and in recent years, many studies emerged in order to protect people from its impact. In fact, it is already well-known that constant exposure to noise will lead to health effects, such as sleep disturbance [52,53], annoyance [54,55], cardiovascular effects [56], learning impairment [57,58], and hypertensive ischemic heart disease [59]. Prevention also depends on understanding temporal patterns of the local noise levels, and thus recent applications use wireless sensor networks for noise monitoring [60,61], representing a modern and cheap solution supporting and augmenting the mandatory noise maps and action plans [62]. In outdoor environments, acoustic barriers are the most widespread solution to mitigate the noise produced by the main sources: railway traffic [63,64], airports [65,66], and industrial plants [67,68]. Recent developments in the field are moving attention towards sonic crystals used as acoustic barriers [69].

We recruited healthy students from Vienna to study the physiological reactions of both cardiovascular and respiratory systems to everyday urban air pollution exposure. We also controlled for noise exposure to disentangle the effects of the two main exposures from road traffic.

## 2. Materials and Methods 

The recruitment of healthy students was performed as part of the diploma thesis of five of the authors (J.P., L.U., E.U., B.E., S.P.). Each student approached friends and colleagues and organized the walks, for a total of 3 to 6 participants each (including him- or herself). These group leaders were responsible for the logistics and data collection. According to the principles and rules of diploma thesis projects at the Medical University of Vienna, each group leader had to apply separately for the ethical approval for his/her project. All ethical approvals were granted by the Ethics Committee of the Medical University of Vienna. Prior to the thesis projects, an overall ethical approval was also obtained from the Ethics Committee of the City of Vienna (EK 15-259-VK_NZ, November 19th, 2015).

In total, twenty-four healthy students walked at least 4 times for one hour under each of the 4 settings: Along a busy road, along a busy road wearing ear plugs, in a park, and in a park but exposed to traffic noise (65 dB) through on-ear headphones with recorded road traffic noise. The road-walk was along the “Hernalser Gürtel”, a main road in Vienna with 400–500 cars/15 min throughout the day (6 a.m.–7 p.m.). The selected park was the “Augarten”, a large park in the center of Vienna. Walks were planned to be scheduled at fixed times of the day, but due to the time constraints of the participants, this was not always possible.

The total mass of particles (PM) with an aerodynamic diameter above 2.5 µm (PM_2.5_), above 1 µm (PM_1_), particle number, and noise levels were measured throughout each walk. PM_2.5_ was measured using a Grimm Portable Laser Aerosol Spectrometer Model 1.108 (Grimm Aerosol Technik, Ainring, Germany). To measure particle numbers (PN), a miniature diffusion size classifier (miniDiSC, http://www.fierz.ch/minidisc/, Institut für Sensorik und Elektronik, Brugg-Windisch, Switzerland) was used. Concentrations were averaged at every 6 s intervals and stored. The noise was measured with a Brühl & Kjaer sound level meter, type 2236 (Brühl & Kjaer, Bremen, Germany). Every 15 min, the equivalent continuous sound level was recorded.

Both spirometric lung function and exhaled nitrogen oxide (NO) were measured before, immediately after, one hour after, and approximately 24 h after each walk. Spirometry was performed using an EasyOne™ Spirometer (ndd Medizintechnik AG, Zürich, Switzerland) in an upright standing position and applying a nose clip following standard procedures [70,71]. NO in exhaled air [72] was measured using the portable instrument NObreath™ (Bedfont Technical Instruments Ltd., Harrietsham, UK).

Blood pressure and heart rate variability (HRV) were measured every 15 min during each walk. For recording heart-rate variability the mobile ECG device eMotion Faros™ (Biomation, Almonte, Ontario, Canada) was used. ECG-files were analyzed over time windows of 15 min each with the Kubios software version 2.2. (Kubios, Kuopio, Finland)

The temperature was obtained from a nearby stationary meteorological station. The data on fine particle (PM_10_) background concentration were also obtained from a nearby fixed monitor operated by the City of Vienna (Station near the General Hospital: AKH).

In panel studies, each participant serves as his or her control. In repeated measurements, time-varying exposures and health indicators are assessed and correlated with each other. Theoretically a study could be performed with a single participant. The power of the study not only depends on the number of participants, but also on the number of observations (time points times participants). With a sufficient number of observations even in a single participant, the effects of exposure can be demonstrated. In this case, it would not be clear if the single participant is representative for a broader population or if, by chance, that participant is for example representative for a highly susceptible subgroup. With multiple participants it is possible to examine if effect estimates differ significantly between them. All statistical calculations were performed with STATA SE Vers. 13.1 (StataCorp LLC, College Station, Texas, USA). We applied the xtreg command that fits regression models to panel data. In particular, xtreg with the fe option fits fixed-effects models; and with the re option, it fits the random effects models by using the Generalized least squares (GLS) estimator. When fixed effect and random effect models provide similar estimates (checked with the Hausman test), significant variation in susceptibility can be ruled out.

The date of respiratory and cardiovascular markers were recorded in two separate files for further statistical analysis. Respiratory parameters (immediately, 1 h and approximately 24 h after the exposure) were assessed for each participant with single air pollution markers serving as independent variables in a random effects GLS regression. Fixed effect models generally provided very similar results, although the Hausman test was significant in few but not in most models. The random effects option is the default setting in STATA and does provide broader confidence intervals than the fixed effects model. Therefore, for the sake of internal coherency, random effects were applied in all models. In two adjusted models, either the same respiratory marker or NO in exhaled air before the exposure were included to control for unmeasured influences prior to the walk.

Cardiovascular markers were assessed every 15 min separately and applied to a random effects GLS regression. Exposures in the preceding 15 min (noise and dust) served as independent variables. The temperature was included in the models as a confounder.

## 3. Results

### 3.1. Participants and Exposure

In a first run from December 2016 to May 2017, 20 students (11 male, 9 female) with an average age of 24 years (range 21–33) walked on average 12 times for one hour (range 8–20). All of them were non-smokers and reportedly healthy. In a second run, four additional (female) students were recruited, who performed their walks in May and June 2018.

Air pollution and ambient noise were determined to be higher near the street but were temporally not correlated with each other. Individual exposure to noise was further de-coupled from ambient conditions by design (ear-plugs and head-phones respectively). Particle measures were found to be well correlated (*R*-values for all particle mass measures including PM_10_ at nearby fixed monitoring station >0.9). Personal exposure concentrations were actually found to be higher than concentrations at the fixed monitoring station, either indicating that the Aerosol Spectrometer overestimated particle mass concentrations systematically, or demonstrating higher personal exposure compared to the fixed station. As an example, hourly values of PM_2.5_ measured with the spectrometer and PM_10_ measured at the fixed station displayed a correlation coefficient of 0.96. A linear regression model with PM_10_ at the fixed station and the setting (road versus park) explained 92.5% of the variation of PM_2.5_. In this model, the difference between road and park was small (4.2 µg/m^3^) but significant (*p* = 0.016). The slope of the regression line (ß of PM_10_ at the fixed station) was 1.57 (*p* < 0.001). Table 1 describes the range of exposure for PM_10_ at the fixed station and the personal exposure measured as PM_2.5_, PM_1_, and PN. Because of the high correlation between the particle mass values, only PM_1_ from the personal monitoring and PM_10_ from the fixed site (controlling for setting—road vs. park) were further analyzed for health effects. In addition, effects of personal particle number concentrations (*R* with mass concentrations between 0.72 and 0.77) were investigated.

After controlling for seasonal and daily trends, ambient noise levels at the road were approximately 10 dB louder than in the park. These ambient noise levels were partly overruled by earplugs and/or headphones. The average sound pressure level L_A,eq_ was ~56 dB at the road. At the road wearing earplugs, noise was assumed to be 30 dB lower. In the park it was measured at about 46 dB and the headphones were set to 65 dB.

### 3.2. Air Pollution and Respiratory Health

Higher air pollution levels were found to be correlated with reduced lung function parameters. Measures of large airway resistance, especially peak-flow (PEF), were in the majority not significantly affected by air particle mass or count, while measures indicative of the small airways like mid-expiratory flow at 25% lung volume (MEF25) showed more consistent effects and remained low even 24 h after exposure. Figure 1 provides two exemplary figures for the generally observed pattern. A more complete overview of the results is provided in Appendix A (Table A1). Measured PM levels (either from the personal monitor or from the fixed site) had a stronger and clearer effect than the setting (road vs. park) and the effect of the setting (“road”) usually lasted less long than the effect of the pollutant concentration. This was expected, because the impact of the setting was restricted to a single hour, while the measured pollution concentration was fairly representative for the whole day and for the whole area.

Exhaled NO levels were found to be significantly reduced immediately after and 1 after the walk with increased dust levels. After controlling for background dust levels, walking besides the road increased exhaled NO levels measured 24 h later (Figure 2). The background reflects the average exposure at that time for a longer period (and thus also affects NO before the setting) while the setting may add to the background exposure for a short and defined period of 1 h. Therefore the effects of pollutant concentration (Figure 2 depicts particle number as an example) were controlled for pre-walk NO levels.

### 3.3. Air Pollution and Noise Effects on Cardiovascular System

Noise levels were found to correlate with reduced blood pressure (stronger effects for systolic than for diastolic blood pressure) and lower heart rate variability. The effects on heart rate variability generally attenuated in the course of the walk, indicating some adaptive process. The temperature also had clear effects on cardiovascular parameters, but did not confound noise effects. Air pollution effects were less pronounced and not very consistent.

Again, only examples of the recorded data are visually presented here (Figure 3) that are typical and representative for effects on other parameters as well. For a more complete overview of the results the reader is referred to Appendix A (Table A2).

## 4. Discussion

Total exhaled NO originates from different exogenous and endogenous sources [73,74,75]. NO is secreted by epithelial cells inducing relaxation of the smooth muscle cells of the bronchial walls. This secretion is inducible and a reaction after a reflective muscular narrowing of the airways: An irritant stimulus will at first lead to increased muscle tone and thus after a very short interval to an increased airway resistance. The increased tonus of airway smooth muscles is then typically quickly antagonized by an increased cellular level of NO, which thus induces a lower airway resistance. NO also serves as messenger molecule in inflammatory processes. Here, it is mostly involved in eosinophilic inflammation and signifies allergic asthma (and thus an increased bronchial reactivity as well as a likely increased airway resistance). Epithelial cells capable of producing NO may be compromised during (neutrophil) inflammation and by some toxic substances (e.g., in cigarette smoke). Therefore, smokers generally have lower NO levels, and after the smoking of cigarettes, NO is further reduced [76,77,78].

NO might relax smooth muscles and thus reduces airway resistance. However, NO in the tissue may also act as an oxidative stressor and thus contributes to an inflammatory response after it is induced by a reflective muscular contraction elicited by an irritant. Therefore, after a certain delay, higher NO levels might predict increased airway resistance again.

These different complex pathways make the interpretation of NO in a panel study a challenge. In this context, timing is essential and even with our protocol of repeated measures (before, immediately after, 1 h after, and 24 h after a defined exposure of one hour) we were not fully able to capture the complete time course. Considering both background pollution levels that are likely to represent the longer term average exposure and the effects per setting which by design lasted only for one hour, we could further disentangle the temporal variation. In this endeavor we were hindered by the finding that differences between settings were small compared to variation in background exposure over time.

Nevertheless, we were able to demonstrate reduced NO levels due to higher background exposures, indicative of toxic damage to epithelial cells, as well as increased levels of NO with 24 h latency after a 1 h acute exposure, indicative of inflammatory response. A reduced lung function led to an increase in NO levels, and higher NO levels were indicative of increased lung function levels at the next measurement point (data not shown).

We could not show consistent effects of the specific setting (road vs. park) on lung function. This could be due to a lack of power because a binary exposure variable provides much less information than a continuous one. We had designed our study following the London study [19] where a panel of 60 asthmatics walked for 2 h, alternatingly along Oxford Street or through the nearby Hyde Park. In that study, significant differences between the two settings could be demonstrated. Asthmatics might be more susceptible to air pollution than healthy adults. The London study examined substantially more participants. The authors also point out that Oxford Street is a street canyon with very high diesel traffic. In Vienna, the particle concentration did differ between the two settings. However, especially regarding particle mass, the daily variation was much more pronounced than differences between the settings. This might be another reason we did not observe consistent effects of the setting.

Unexpectedly, the only clear and significant effect of the setting was a higher peak flow (PEF) after the walk on the road. The peak flow is sensitive of the participant’s cooperation and muscular efforts. Keeping in mind that the participants were not blinded to the setting, the higher PEF values after the walk along the road could be due to the participants’ subconscious ambition to try harder.

Also with measured air pollutants effects on the larger airways, as signified by PEF, the expected indicative of a muscular reflex response could not be demonstrated consistently. Effects on the small airways signified (e.g., by MEF25) persisted for 24 h and likely represent inflammatory responses that might be clinically more relevant.

Lung function values mostly remained in the normal physiological range. This is not surprising, considering the comparatively low exposures and the generally healthy state of the participants. We observed effects during everyday activities in everyday settings and therefore did not expect to observe severe detrimental health effects. However, it seems noteworthy that even under these conditions and even in young healthy adults with a comparatively small number of subjects and a moderate number of repeated observations, several relevant effects were reached. While acute effects shortly after the exposure are likely due to physiological protective reactions under nervous control, the longer term changes, as most clearly seen in the end expiratory flows, are indicative of an increase in resistance in the small airways and are likely caused by inflammatory tissue responses. This is a cause of concern, as even small effects, when cumulated over the years, will in the long run hasten the functional decline of the respiratory system.

Cardiovascular effects were mostly observed in relation to noise levels. These effects remained even after controlling for temperature, which was the most important predictor of heart rate, heart rate variability, and blood pressure. Also these effects remained in the physiological range, which again was to be expected. Even subtle effects to everyday exposures during everyday activities still can be of concern in the long run.

Health effects of environmental noise exposure are well established [79,80,81,82,83,84,85,86]. However, effects of noise on annoyance are usually assessed through cross-sectional studies [83] and effects on cardiovascular health through cohort and case-control studies [86]. Acute reversible physiological effects of noise have not been investigated that much lately. Effects of much higher occupational noise levels have been studied in relation to sleep [87] stress hormones [88,89,90], or blood pressure [91]. However, studies on acute effects of environmental noise exposure are rare [92]. The latter study was similar to ours as it also examined the effects of air pollution and noise in a panel of healthy young adults (33 men and 33 women). Air quality was assessed based on available stationary data while noise exposure was measured by personal monitoring. This study investigated blood pressure as the only outcome. Similar to our study air pollution led to higher blood pressure but contrary to our findings also noise increased blood pressure.

Other acute effects of air pollution have already been demonstrated in some panel studies, notably on glucose metabolism [93,94] and inflammation [95]. However, we are confident that we have covered the most important endpoints, including respiratory [96] and cardiovascular [92,97] effects. The observed respiratory effects of this study are generally in line with those reported in the literature. The two studies most similar to our approach [50,51] investigated a small group of cyclists (15 and 12 respectively) riding on low- and high-traffic intensity routes. Both studies examined lung function only by comparing settings. The former did not find significant differences while the latter unexpectedly found a better lung function immediately after the ride on the high-intensity route, but poorer lung function 6 h later. Their results are reflected by our study, as we also failed to find consistent effects per setting and we could also demonstrate lasting effects.

Cardiovascular effects of air pollutants are typically reported in patients and not in healthy young adults. Only one study [47] examined HRV in healthy adults, but this study monitored HRV (over 5 min intervals) during sleep only. A clearly vegetative regulation of cardiovascular function will differ between sleeping and waking hours. In our study, the only observed effects of air pollution consisted of a small increase in blood pressure.

Our study design did not allow for the examination of lag periods for cardiovascular effects. This might also explain why cardiovascular effects were predominantly seen with noise and not with air pollution. Traffic noise is considered an important environmental stressor. Because of this, we would have expected a positive association between sound pressure level and blood pressure. Interestingly, the reverse was seen in our study while the reduction in HRV confirms our hypothesis.

## 5. Conclusions

Exposures to air pollutants cause statistically significant respiratory reactions, even in healthy young adults. These effects were visible even with a relatively small number of participants and only few repeated measurements. The effects were reversible and generally not very severe, but nevertheless clear and consistent. This was especially true for measured air pollutants that were not only representative of the exposure of the single hour of the walk at the road or in the park, but also of the general air quality on that day and in the whole region. Noise had a clear effect on most cardiovascular parameters. For most of the endpoints, the noise effects displayed attenuation over the course of the walk. Contrary to expectations, higher noise levels lead to lower blood pressure.

We cannot fully exclude the possibility that the latter unexpected findings are due to a lack of study power. Indeed, in a first analysis of the incomplete data set, noise effects on blood pressure were even stronger. However, unmeasured confounding seems to be a more likely explanation, although controlling for temperature as a surrogate of seasonal and diurnal variation did not strongly affect our findings. In addition, we were not able to show effects of setting (road compared to park), and this is likely due to poor power because of the small sample size.

Epidemiological studies often focus on easy-to-measure lung function parameters like FVC, FEV1, FEV1/FVC, or PEF. However, higher resistance in the small airways is better reflected by end expiratory flows (MEF50, MEF25). These measures showed the clearest results and the longest lasting effects. While immediate reductions in lung function could be interpreted as signs of protective reflexes of the bronchial muscles, prolonged reductions in the flow values might be due to an inflammatory swelling of the mucosa, and thus would be clearly an indicator of an adverse effect.

## Figures and Tables

**Figure 1 ijerph-16-00788-f001:**
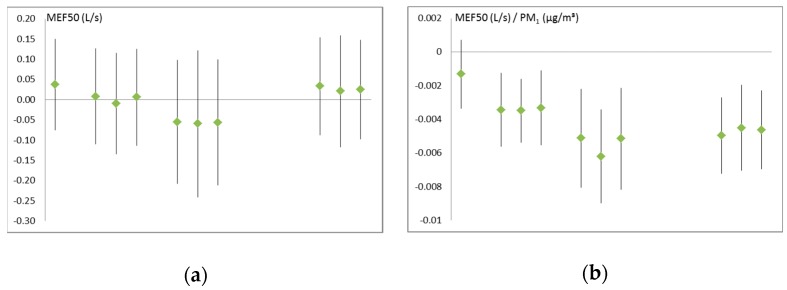
Examples for air pollution effects on MEF50* (in L/s): (**a**) Effect of the setting “road”, controlled for background pollution; (**b**) Effect of PM_1_ (µg/m³). Each triplet stands for: unadjusted, adjusted for lung function before exposure, adjusted for exhaled nitrogen oxide (NO) before exposure. The time points are (left to right): Before exposure (one value only), immediately after the walk, 1 h after the walk, and 24 h after the walk. * For Abbreviations please see Table A3!

**Figure 2 ijerph-16-00788-f002:**
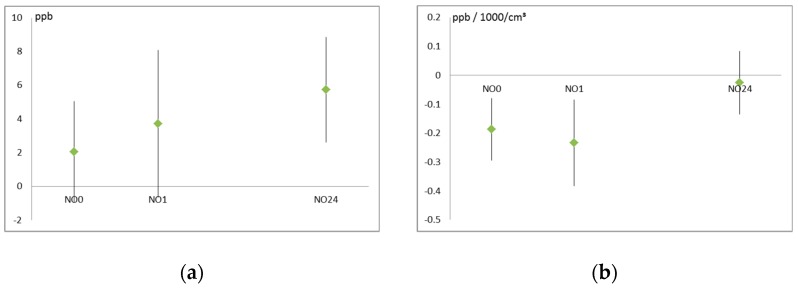
Effect of exposure on exhaled nitrogen oxide (NO) values (in ppb) at different time points (immediately after the walk, 1 h after the walk, and 24 h after the walk): (**a**) Exposure near the road compared to a park, controlled for background concentration; (**b**) Effect of particle number (per 1000 particles per cm³) controlled for setting.

**Figure 3 ijerph-16-00788-f003:**
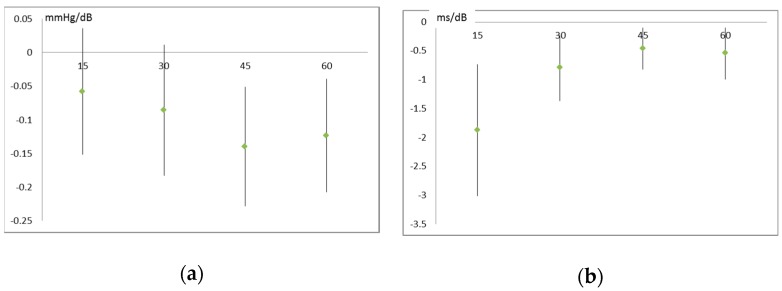
Effect of noise on cardiovascular parameters: (**a**) systolic blood pressure; (**b**) Standard deviation of intervals between two consecutive heartbeats. Noise levels are averaged over 15 min (4 periods per 1 h walk) each. Blood pressure was measured at the end of each period; electrocardiogram readings were taken over the whole 15 min period.

**Table 1 ijerph-16-00788-t001:** Exposure to particles during walks.

Metric	Arithmetic Mean	± Standard Deviation	Range
PM_10_ fixed station	28.0 µg/m³	26.5	5–95
PM_2.5_ personal	38.7 µg/m³	43.5	2–146
PM_1_ personal	31.0 µg/m³	38.9	1–133
PN personal	21,347.8/cm³	18,826.5	41,989–80,0596

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
