# Peer review of "Acute Effects of Air Pollution and Noise from Road Traffic in a Panel of Young Healthy Adults"

_ijerph, 2019, doi:10.3390/ijerph16050788_

Round 1
Reviewer 1 Report
The paper quality is undoubtedly low from a lot of point of view, thus I believe the authors should work a lot on the paper before submitting again to the journal. The most important aspects to change are from a methodology point of view, writing style and organization of the text.
At first, proposing an epidemiological study with just 24 cases is very ambitious and optimistic from the authors’ point of view. Thus, please consistently increase the subjects.
Then, the authors should definitely spend more time in writing the paper. Do they ever have seen a paper with only 6 lines of introduction? With such a short abstract? Also some periods are put in multiple times in the text.
Reference are inserted in a confusing way in the text. Sometimes even 15 references are put together. Abstract, introduction, references and conclusion are sections extremely important to allow people understand the paper and promoting your research.
English writing style is also weird and it appears to evident that some authors wrote a chapter while others wrote different chapter. Please uniform all.
Author Response
see first page of the document "response_ijerph-434769"!

Reviewer 2 Report
The manuscript presents an interesting study on the cardiopulmonary impacts of exposure to particle (both in terms of mass and number concentrations) and noise pollution in 24 adults aged 21-33 years old.
The study used a very small sample size and the whole measurement period was 1 hour for each participant. Therefore this work can be considered as a pilot study only. Limitations and uncertainties in the resulting outcomes should be acknowledged in methods sections as well as conclusions.
The use of “young healthy subjects” in the title and subsequent descriptions up to the results section abstract imply that the participants were not adults (but children). This is misleading and for clarity, the title must include the word “adults”. This is because the health impacts in children and adults are not in the same scale.
Literature review and citations need revision.
Please add more descriptions on the methods used for data collection and interpretation.
How the participants were recruited? what were the inclusion criteria, including the age range for the participants to be included?
Please include details of the ethical approval for recruiting the participants.
I would provide a full review of the manuscript if these details were provided.
Author Response
see page 2 of the document "response_ijerph-434769"!

Round 2
Reviewer 1 Report
After the previous round of revision, i can report that now the paper’s quality is finally decent, thus deserving a more detailed round of revision. However, there are still some point to be fixed before thinking of a publication.
· At first, many references have been added, but it now seems that authors are more expert about air pollution than noise exposure. In fact, even if the title says so, noise exposure is firstly introduced at the end of the introduction, while it deserves more attention. I would help the authors providing a period that would improve its introduction while completing the references section from a noise point of view:
Noise pollution is a major concern for modern world and in recent years many studies emerged in order to prevent people for being impacted by it. In fact, it is already well-known that constant exposure to noise will lead to health effects such as sleep disturbance [Muzet, A. (2007). Environmental noise, sleep and health. Sleep medicine reviews, 11(2), 135-142. https://doi.org/10.1016/j.smrv.2006.09.001 ; de Kluizenaar, Y., Janssen, S. A., van Lenthe, F. J., Miedema, H. M., & Mackenbach, J. P. (2009). Long-term road traffic noise exposure is associated with an increase in morning tiredness. The Journal of the Acoustical Society of America, 126(2), 626-633. https://doi.org/10.1121/1.3158834], annoyance [Miedema, H. M., & Oudshoorn, C. G. (2001). Annoyance from transportation noise: relationships with exposure metrics DNL and DENL and their confidence intervals. Environmental health perspectives, 109(4), 409. doi:10.1289/ehp.01109409; Fredianelli, L., Carpita, S., & Licitra, G. (2019). A procedure for deriving wind turbine noise limits by taking into account annoyance. Science of the Total Environment, 648, 728-736.], cardiovascular effects [Babisch, W., Beule, B., Schust, M., Kersten, N., & Ising, H. (2005). Traffic noise and risk of myocardial infarction. Epidemiology, 33-40. DOI: 10.1097/01.ede.0000147104.84424.24], learning impairment [Lercher, P., Evans, G. W., & Meis, M. (2003). Ambient noise and cognitive processes among primary schoolchildren. Environment and Behavior, 35(6), 725-735. https://doi.org/10.1177/0013916503256260; Chetoni, M., et al. (2016). Global noise score indicator for classroom evaluation of acoustic performances in LIFE GIOCONDA project. Noise Mapping, 3(1). DOI 10.1515/noise-2016-001] and hypertension ischemic heart disease [Van Kempen, E., & Babisch, W. (2012). The quantitative relationship between road traffic noise and hypertension: a meta-analysis. Journal of hypertension, 30(6), 1075-1086. doi: 10.1097/HJH.0b013e328352ac54]. Prevention include understanding, even in real time, the noise levels in the area and thus recent application consist in using wireless sensor network for noise monitoring [Zambon, G., Roman, H., Smiraglia, M., & Benocci, R. (2018). Monitoring and prediction of traffic noise in large urban areas. Applied Sciences, 8(2), 251; Zambon, G., Benocci, R., Bisceglie, A., Roman, H. E., & Bellucci, P. (2017). The LIFE DYNAMAP project: Towards a procedure for dynamic noise mapping in urban areas. Applied Acoustics, 124, 52-60], representing a modern and cheap solution to fulfil with the mandatory noise maps and action plans [Licitra, G., et al. (2017). Prioritizing Process in Action Plans: a Review of Approaches. Current Pollution Reports, 3(2), 151-161. https://doi.org/10.1007/s40726-017-0057-5]. In outdoor environment, acoustic barriers are the most widespread solution to mitigate the noise produced by the main sources: railway traffic (Licitra, Gaetano, et al. "Annoyance evaluation due to overall railway noise and vibration in Pisa urban areas." Science of the total environment 568 (2016): 1315-1325. Bunn, Fernando, and Paulo Henrique Trombetta Zannin. "Assessment of railway noise in an urban setting." Applied Acoustics 104 (2016): 16-23), airports (Gagliardi, Paolo, et al. "ADS-B System as a Useful Tool for Testing and Redrawing Noise Management Strategies at Pisa Airport." Acta Acustica united with Acustica 103.4 (2017): 543-551. Iglesias-Merchan, Carlos, Luis Diaz-Balteiro, and Mario Soliño. "Transportation planning and quiet natural areas preservation: Aircraft overflights noise assessment in a National Park." Transportation Research Part D: Transport and Environment 41 (2015): 1-12.), industrial plants (Kephalopoulos, Stylianos, et al. "Advances in the development of common noise assessment methods in Europe: The CNOSSOS-EU framework for strategic environmental noise mapping." Science of the Total Environment 482 (2014): 400-410. Morel, Julien, Catherine Marquis-Favre, and L-A. Gille. "Noise annoyance assessment of various urban road vehicle pass-by noises in isolation and combined with industrial noise: A laboratory study." Applied Acoustics 101 (2016): 47-57.). Recent developments in the field are moving the attention towards sonic crystals used as acoustic barriers [Fredianelli, L. et al. (2019). Recent Developments in Sonic Crystals as Barriers for Road Traffic Noise Mitigation. Environments 2019, 6(2), 14; https://doi.org/10.3390/environments6020014].
· Simultaneously, if the authors believe that the total references will be too many (which in my opinion in never a bad idea), the authors can remove some reference when they are still too many put together under the same bracket (e.g. [1-6], [9-14], and so on).
· Graphs are badly reported. Units should be reported in the scale and not in the title, scale should be the same for similar figures.
· Conclusions are short and poor. Please improve them.
Author Response
see document "response 2_1"!

Reviewer 2 Report
Generally, the manuscript has improved. Some minor comments are as follows:
Abstract and conclusion must include an statement on limitations of the study and uncertainties, including the very small sample size.
Section 2. Materials and Methods: please add if all the 24 participants walked along the same roads and park. If not, please add details traffic characteristics of the roads. It s stated that they walked along busy roads. How and on what basis a road was defined as “busy”? When they walked wearing an earplug, did they walk along the same busy road that they walked without the earplug? Did each participant walk with and without earplugs at the same time of the day? Were the walks (for all participants) done at the same time of the day? Time of the day affects the traffic characteristics, hence the exposures and also the physiological responses. If not please add these as sources of uncertainties.
3.1. Participants and exposure: results are averaged over all participants. Since data collection was done over different seasons, please include seasonal variations or if they were statistically significant. Was there any difference between the results for female and male participants?
section 3.1. Participants and exposure states that “Personal exposure concentrations were actually found to be higher than concentrations at the fixed monitoring station indicating that the Aerosol Spectrometer overestimated particle mass concentrations systematically.” This is not correct by default and such a statement cannot be made based on these data. Personal exposures are often higher than data measured at fixed sites, especially if the personal data were collected near roads/busy roads. Also, there is no clear-cut linear correlation between PM10, PM2.5 and PM1 and the relationships highly depends on the sources of the particles. Please revise.
Figures 1 and 2: instead of examples, please present an overall visual presentation of the data set.
Author Response
see document "response 2_2"!

Round 3
Reviewer 1 Report
The authors followed all my suggestions, thus i believe the paper now has reach the sufficient quality for being published. I also wish to congratulate with the authors for the success in this process.